

# Assessing the performance of various fire weather indices for wildfire occurrence in Northern Switzerland

Daniel Steinfeld[1], Adrian Peter[2], Olivia Martius[1], and Stefan Brönnimann[1]

[1]Institute of Geography and Oeschger Centre for Climate Change Research, University of Bern, Switzerland
[2]Office for Forests and Natural Hazards of the Canton of Bern

**Correspondence:** Daniel Steinfeld (daniel.steinfeld@giub.unibe.ch)

**Abstract.** Fire weather indices are widely used to understand and assess meteorological fire hazard. However, in complex regions such as Switzerland with mountainous and hilly terrain, it is difficult to select an appropriate index. In this study, we validate the performance of 14 fire weather indices, four meteorological variables, and a logistic regression model to predict wildfire occurrence for different ecoregions in the canton of Bern in Northern Switzerland with respect to historical fire records

from 1981 to 2020. We find that the performance of the indices varies seasonally and regionally. The spring season (March-May) shows that the Canadian Fine Fuel Moisture Content and other indices that respond readily to weather changes perform best. In summer (June-August) and autumn (September-November), the Canadian Buildup Index and Drought Code - indices that describe persistent hot and dry conditions - perform best. Overall, seasonal differences in performance are larger than inter-regional differences. Finally, we show that a logistic regression model trained on local historical fire activity can outperform

existing fire weather indices and can be used for medium-range weather forecasting or climate change studies, using only daily averages of meteorological variables as input.

## 1 Introduction

Fire weather indices are commonly used by fire management agencies to asses and predict weather conditions that are most conducive to wildfire. Such fire weather indices provide a measure of the daily fire danger, i.e., the risk of fire occurrence, but

also of the expected fire intensity and rate of spread. They are calculated based on meteorological information from weather stations or numerical weather prediction models. Fire danger is related to soil and vegetation dryness (fuel moisture, see, e.g., Thornthwaite, 1948; Nesterov, 1949; Käse, 1969), which in turn is directly related to temperature, precipitation and humidity. Wind speed is a factor influencing the rate of fire spread (Reinhard et al., 2005; Potter and Potter, 2012). Therefore, fire danger is high in hot, dry and windy weather.

A number of different fire weather indices exist and are currently used in operational warning systems (e.g., Giuseppe et al., 2016). Some indices are based on very simple algorithms that combine temperature and humidity information (Sharples et al., 2009), while others are more complex and estimate both fire occurrence and severity (Van Wagner and Pickett, 1985). The Canadian forest fire weather index system (CFFWIS, Van Wagner and Pickett, 1985; Wotton et al., 2009) consists of several sub-indices. It considers the effects of weather on fuel moisture and fire behavior. The CFFWIS is the most widely used



fire weather index globally, both in practice and in research. It is used to predict fire danger in several European countries, including the European Forest Fire Information System (EFFIS López et al., 2002), although it was originally developed for use in Canadian pine forests.

While Dimitrakopoulos et al. (2010); Wastl et al. (2012); Gudmundsson et al. (2014); Abatzoglou et al. (2018) and many others have identified weather as a primary driver of regional fire activity, fire occurrence is not solely dependent on weather
conditions, but rather the result of complex interactions among weather, fuel availability and dryness, topography and human activities (e.g., Schumacher and Bugmann, 2006; Weibel, 2009). Indeed, the vast majority of fires are human-caused (DeWilde and Chapin, 2006; Zumbrunnen et al., 2011; Pezzatti et al., 2013; Syphard et al., 2017) in contrast to naturally caused fires, which are almost exclusively caused by lightning strikes (Conedera et al., 2006; Moris et al., 2020). Because fire weather indices are based solely on meteorological information, they cannot provide perfect prediction of fire occurrence, but rather
measure antecedent conditions (Andela et al., 2017). Furthermore, they are based on empirically derived correlations between weather and fire for specific climatic and vegetation conditions. This means that the transferability of the indices to other regions and under changing climatic conditions is limited (Weibel et al., 2010; Reineking et al., 2010; Krawchuk and Moritz, 2011), and their application outside the "area of origin" requires careful evaluation and adaptation (Weibel, 2009; Wotton et al., 2009; Padilla et al., 2011; de Jong et al., 2016; Bekar et al., 2020). In this paper we assess the performance of different fire
weather indices for Northern Switzerland

It is difficult to find an appropriate fire weather index, especially for regions with complex and diverse terrain like Switzerland. The importance of meteorological, ecological and anthropogenic factors (e.g., regulation at the municipal level, fire bans) affecting fire occurrence can vary greatly at relatively small geographic scales (Reineking et al., 2010; Conedera et al., 2018; Bekar et al., 2020), and the predictive value of individual indices varies greatly depending on the region and season to which
they are applied (Padilla et al., 2011; Arpaci et al., 2013; Angelis et al., 2015; de Jong et al., 2016; Pezzatti et al., 2020). Some researchers have attempted to account for these interactions by using statistical approaches to model the local relationship between fire activity and meteorological conditions based on local fire statistics (Weibel et al., 2010; Angelis et al., 2015; Erickson et al., 2016). Some statistical models even consider non-climatic variables (e.g., distance to infrastructure Reineking et al., 2010; Vacchiano et al., 2018). Angelis et al. (2015) combined several fire weather indices in a "fire niche" approach
based on machine learning to better predict the occurrence of fire in a given region, which was successfully implemented for operational use in Southern Switzerland. Here, the focus is on Northern Switzerland.

Switzerland, located in the mountain ranges of the Central European Alps, is now considered a country with low to moderate forest fire frequency (Conedera and Tinner, 2010), although the more continental areas, e.g., the dry Alpine valleys and the southern canton of Ticino, have a significant fire risk under current climatic and land use conditions (Zumbrunnen et al.,
2012). According to the Swiss Federal Institute for Forest, Snow and Landscape Research (WSL), close to 100 forest fires are registered annually in Switzerland, covering a total area of about 300 hectares (mean values from 1990 to 2014, Pezzatti et al., 2016). In Switzerland, forest fires rarely cause damage to buildings or directly endanger human lives. However, at a local scale, alpine forest fires can have negative effects on the forest ecosystem and the protective function of forests against avalanches (Conedera et al., 2003; Brang et al., 2006; Rickli and Graf, 2009). In Switzerland, temperatures have increased almost twice as





much as the global average. This increase is consistent with climate change projections, which identify the Swiss Alps as an area particularly sensitive to the effects of global warming (NCCS, 2021), and further warming is associated with a 10 to 30% decrease in summer precipitation (Remund et al., 2016; Sørland et al., 2020), earlier snowmelt and a decrease in snow cover in winter (Vorkauf et al., 2021) and more severe or prolonged heat waves and droughts (NCCS, 2021). These future changes contribute to weather conditions that increase fire potential (Cane et al., 2013; Flannigan et al., 2016), and as a direct result, wildfire frequency and intensity are likely to increase even in areas of Switzerland and seasons previously classified as not at risk of fire (Schumacher and Bugmann, 2006; Dupuy et al., 2020). Based on climate scenarios for Switzerland (NCCS, 2021), the danger of more and larger fires could also increase in the future north of the Alps (Pezzatti et al., 2016), a region with little published research regarding index-based assessment of wildfire, and operationally used fire weather indices have not been explicitly validated and verified for Northern Switzerland.

In this study, we evaluate several fire weather indices, meteorological variables and a logistic regression model for their ability to predict fire-prone weather conditions and fire occurrence. The aim is to identify the best performing index for Northern Switzerland. The study is conducted under the Wyss Academy for Nature project "Forest Fire management on the northern side of the Alps", which aims to aid the Office for Forests and Natural Hazards of the canton of Bern in creating a comprehensive risk management system adapted to climate change for the prevention and control of forest fires (Peter and Pfammatter, 2019). The best performing index will serve as the basis for hazard assessment and risk classification for the entire canton and the northern side of the Alps. Due to the high complexity of the Swiss landscape with mountainous terrain, we apply the validation separately for the three biogeographical ecoregions on the northern side of the Swiss Alps. We draw on experiences and findings from regions on the southern side of the Alps (e.g., Angelis et al., 2015), and adapt them to the specific conditions of Northern Switzerland. The paper is structured as follows: The datasets, methods and derived fire weather indices are specified in Section 2. Section 3 presents the index-based validation, and the results are discussed in detail in Section 4. The last Section 5 provides a summary and some conclusions.

## 2 Data and Methods

### 2.1 Study area and Fire Database

The study area is the canton of Bern, which is located in the northern part of Switzerland (see Figure 1) and covers an area of 5959 km$^2$ with an elevation range of 400 to over 4000 m asl. It is the second largest of the Swiss cantons and represents all three distinct and diversified biogeographical ecoregions found in Northern Switzerland (Gonseth et al., 2001): the high mountain ranges and deep valleys of the northern Pre-Alps (northern Alps), the low and narrow mountain ranges of the Jura (Jura), and the hilly and densely populated Central Plateau in-between (Plateau).

The climate in the study area is mild and humid, with warm, wet summers (17.5 °C JJA average) and cool to mild winters (0 °C DJF average), and is mainly determined by westerly winds from the Atlantic Ocean and the perpetual passage of high and low pressure systems (Kottek et al., 2006). The largest seasonal mean precipitation north of the Alps usually occurs in summer, when dry periods alternate with short intervals of convective precipitation (Isotta et al., 2014). The annual mean temperature



and precipitation sum is similar in the three ecoregions, ranging from ~1000 mm at low elevations to ~1400 mm at higher elevations. Related to the complex topography, local wind systems can form (MeteoSwiss, 2015) and play an important role in

drying out the fuel or the spread of wildfire (Sharples et al., 2010; Wastl et al., 2013). Two well-known examples are the foehn, a wind that crosses the main Alpine ridge and leads to a warm and dry downslope windstorm in the lee (Richner and Hächler, 2013; Sprenger et al., 2016), and the Bise, an easterly wind that is enhanced in the Plateau region by channeling between the Jura and the Alps (Wanner and Furger, 1990; MeteoSwiss, 2015). The canton of Bern has a regionally varying forest cover with an average cover of about 31% (Brändli, Urs-Beat et al., 2020). Coniferous trees (mostly spruce and fir) are the dominant

tree species in the higher elevation areas and broadleaved trees (beech-oak forests) generally dominate in the lower regions and in the valleys of the mountainous regions. The vegetation of the study area is strongly influenced by human activities such as agriculture or tourism.

Based on recorded fires from the national forest fire database "Swissfire" of the WSL Swiss Federal Institute for Forest (Pezzatti et al., 2010, 2019), we use here the historical fire occurrence in Switzerland from 1981 to 2020. For this period, the

forest fire database for the canton of Bern can be considered almost complete thanks to intensive retrospective source analysis (Pezzatti et al., 2016). For each fire record, the database provides information about the geographic location (municipality), the date of ignition, the total burned area, and the cause of the fire. Information on cause ("unknown") and total burned was not available in all records. In addition, the MODIS active fire product was used as a second estimate for 2003–2020 to fill potential gaps in database records (Giglio et al., 2016). The MODIS fire product with daily and 1 km resolution combines

MODIS Thermal Anomalies and the Fire product MCD64a1.006. Data were downloaded for Switzerland from NASA FIRMS (2018). Of the total 1180 entries between 2003 and 2020, 385 fire ignitions were not recorded in the databases. They occurred mainly in the Swiss Plateau, 15 of them in the canton of Bern.

Both forest and grassland fires are considered in this study. No distinction was made between human-caused and lightning-caused fires, since the fraction of lightning-caused fires in the canton of Bern is only 4 %.

**2.2  Weather stations and Fire Weather Indices**

To calculate the fire weather indices, we select a representative weather station from the MeteoSwiss SwissMetNet network for each ecoregion, namely Bern/Zollikofen (BER, 552 m asl) in the Central Plateau, Interlaken (INT, 577 m asl) in the northern Alps and La Chaux-de-Fonds (CDF, 1017 m asl) in the Jura. Figure 1 shows the location of the three weather stations. These stations have a long, homogeneous time series for the last 40 years of all standard meteorological variables. The data include

temperature (T), relative humidity (H), wind speed (U) and 24 h cumulative precipitation (P) for the time period 1981–2020. Several existing and widely used fire weather indices were then calculated on a daily basis, resulting in 14 different fire weather indices (Table 1). The indices include the relatively simple Angstrom index (Chandler et al., 1983), the drought index Nesterov (Nesterov, 1949) and the more complex CFFWIS system with its six sub-indices (Van Wagner and Pickett, 1985). For a detailed description of each method, see the original publications in Table 1 or a summary of the methods provided by WSL:

https://wikifire.wsl.ch/. In addition, we use daily mean temperature (daily T), humidity (daily H) and wind speed (daily U)) and the precipitation sum over the past seven days (WeekRain), to examine the relative importance of the raw meteorological



**Table 1.** Fire indices used in this study and related meteorological input variables (temperature T, relative humidity H, wind U and precipitation P) used for calculations. For more information, see original publications or summary documentation provided by WSL: https://wikifire.wsl.ch/.

| Fire indices | Acronym | Meteorological input | | | | References |
|---|---|---|---|---|---|---|
| | | T | H | U | P | |
| Fine fuel moisture code | FFMC | ● | ● | ● | ● | Van Wagner and Pickett (1985) |
| Duff moisture code | DMC | ● | ● | | ● | Van Wagner and Pickett (1985) |
| Drought Code | DC | ● | | | ● | Van Wagner and Pickett (1985) |
| Initial spread index | ISI | ● | ● | ● | ● | Van Wagner and Pickett (1985) |
| Buildup index | BUI | ● | ● | | ● | Van Wagner and Pickett (1985) |
| Fire weather index | FWI | ● | ● | ● | ● | Van Wagner and Pickett (1985) |
| Keetch-Byram drought index | KBDI | ● | ● | | ● | Keetch and Byram (1968) |
| M68dwd | M68dwd | ● | ● | | ● | Käse (1969) |
| Mc Arthur Mark 5 forest fire danger index | FFDI | ● | ● | ● | ● | McArthur (1967) |
| Sharples | Sharples | ● | ● | ● | | Sharples et al. (2009) |
| Fosberg fire weather index | FFWI | ● | ● | ● | | Fosberg (1978) |
| Angström index | Angström | ● | ● | | | Chandler et al. (1983) |
| Nesterov ignition index | Nesterov | ● | ● | | | Nesterov (1949) |
| Baumgartner index | Baumgartner | ● | | ● | ● | Baumgartner et al. (1967) |

variables and as input for fitting a logistic regression model (logit). The logistic regression (Cox, 1958) is ideally suited for statistical modelling of binary response data such as fire occurrence (e.g., presence-absence (1/0) of fires on a given day) (Garcia et al., 1995; Andrews et al., 2003; Padilla et al., 2011; Angelis et al., 2015). It estimates the probability of fire occurrence using a logit function between daily fire occurrence and the linear combination of the meteorological parameters (daily T, H, U and WeekRain):

$$P(\text{fire day} = 1|x_i) = \frac{1}{1 + exp(-\eta_i)} \tag{1}$$

where $P$ is the probability of fire occurrence on a specific day and $\eta_i = \beta_0 + \sum_{j=1}^{k} \beta_j x_{ji}$ is the linear predictor function of $k$ (here 4) meteorological variables. Because it is a linear model using only daily means as input, it is easy to compute and interpret. The performance of the logit model was evaluated using a 10-fold cross validation in which the area under the Receiver Operating Characteristic curve was calculated (AUC; Hanley and McNeil, 1982). Because the logit model was trained using fire records in Bern, it indirectly accounts for local fire occurrence and other non-meteorological factors (Reineking et al., 2010). We also combined a set of indices in a logistic regression model, similar to Angelis et al. (2015), but the performances were comparable to the logit model with daily means.





It is common to use individual stations to provide an estimate of daily fire danger for large areas (Andrews et al., 2003). Although single weather station observations cannot capture the spatial heterogeneity within a complex topography, they still provide an objective comparison of the performance of different indices under the same conditions.

## 2.3   Validation metric

In certain environments, some indices predict fire danger better than others (Van Wagner and Pickett, 1985), and regional
differences in index performance are evident even at relatively small spatial scales (Arpaci et al., 2013). Consequently, it is important to assess the performance of different fire weather indices for optimal operational use, especially in regions with complex and diverse terrains like Switzerland. While fires occur under different weather conditions, a good index should perform well on fire-prone days, such that index values should be higher on fire days than on non-fire days. Here, we apply the "ranked percentile curve" of Eastaugh et al. (2012) as a validation metric. This metric was applied by Arpaci et al. (2013);
Eastaugh and Hasenauer (2014); de Jong et al. (2016) to asses the performance of various fire weather indices. A brief description of the ranked percentile curve approach can be found here. We use fire records from the Swissfire database to divide days from 1981–2020 into fire days (days when at least one fire ignition was recorded) and non-fire days per region. For each index, the daily values are first converted to percentiles with respect to climatology, and the percentiles of fire days are extracted by ascending rank to create a ranked percentile curve (see Figure 3 in Arpaci et al. (2013)). We then fit a non-parametric regression
model to this curve using the Theil–Sen method (Theil, 1950; Sen, 1968), a median-based approach that is robust to outliers. The intercept from the Theil–Sen model is used to assess and compare the performance of the indices. A perfect index would have an intercept of 1 because the index values on fire days would be at the upper tail ("extreme") of the frequency distribution. The ranked percentile curve is threshold-independent and summarizes the ability of a given fire weather index to correctly distinguish between fire days and non-fire days and to identify extreme fire-prone conditions. We use the intercept as our ranked
percentile score to compare the indices and find the best performing index.

To account for spatio-temporal variability, we perform the validation analysis separately for each ecoregion (Jura, Plateau, northern Alps) and season (MAM, JJA, SON). We subdivide fire records by ecoregion and test the performance of each index against the fires that occurred in each ecoregion (Figure 1). For robustness, we also apply the AUC (Hanley and McNeil, 1982), which has been used previously to assess the performance of fire weather indices (Angelis et al., 2015; Pezzatti et al., 2020).
Results of the 10-fold cross validation (repeated 3 times) can be found in the Supplement Figure S1 and are very similar to the ranked percentile curve and are therefore not further discussed in this study. All statistical analyses were performed using the Python package SciPy (Virtanen et al., 2020).



## 3 Results

### 3.1 Analysis of historical fire records

We begin with an exploratory analysis of the spatial and temporal distributions of historical fire records from the database. The spatial distribution of fire activity in Switzerland (January 1981–December 2020) is shown in Figure 1b. During this period, 5458 fire events were recorded in forests and grasslands throughout Switzerland. It is evident that fires occur in all ecoregions of Switzerland, although the number and size (burned area) of fires is significantly higher in the southern canton of Ticino and in the dry inner Alpine valleys. The Alps act as a climate barrier between the north and south of Switzerland, with the south

being warmer by almost +2 K (annual mean) and under the influence of the Mediterranean Sea. In addition, the inner Alpine valleys have their own climate with relatively dry conditions throughout the year, as they are shielded from precipitation. The occurrence of fires and their causes in Ticino and dry inner valleys have already been studied in detail, and the interested reader is referred to Reineking et al. (2010); Angelis et al. (2015); Bekar et al. (2020). If we turn our focus to the northern side of the Alps, about 37 % of all recorded fires occurred in Switzerland. Between 1981 and 2020, a total of 857 fires occurred in

the canton of Bern, with an average of 22 fires per year, of which 124 occurred in the Jura, 420 in the Plateau, and 313 in the northern Alps (Figure 1). Fires do not occur frequently, and the frequency of fire days (ratio between days with recorded fires and all days between 1981–2020) is about 3%. Most fires in Bern are very small, and 50 % are smaller than $100\,m^2$ (0,01 ha). The network of Bernese fire departments is dense, as is the population in general. Even very small fires are detected early and reported to the authorities, so that the burn areas often remain small. The largest fire recorded in Bern during our study period

affected an area of $150000\,m^2$ (15 ha) in Meiringen (located in the northern Alps) in 1982. 39 % of all records in Bern have no estimated size and can be assumed to be small. In this study, the focus is on the ignition/occurrence of fires, as we assume that fire behaviour is not of great importance for very small fires (Arpaci et al., 2013). Following Angelis et al. (2015), we do not filter out small fires or fires without size information for further analysis.

Figure 2 shows the monthly occurrence of fire events in Bern from 1981–2020. Fire activity is highly seasonal with two

distinct fire peaks for all ecoregions, the first in spring (March to April) and the second during summer (July to August), and with limited fire occurrence in the cold season (December to February). Lightning-caused fires typically occur during summer and affect coniferous forests at high altitudes and on steep slopes of the Alps (see also Conedera et al., 2006; Moris et al., 2020). Fires with an unknown ignition source are probably human-caused (Müller et al., 2013; Moris et al., 2020). The spring fire season, when precipitation is low and leaf unfolding has not yet started, is characterized by rapidly spreading surface

human-caused fires, half of which are caused by negligence. The fires that occur in summer are usually smaller and happen with low fuel moisture and hot and dry weather that promote high-intensity fires particularly in coniferous forests (Conedera et al., 2006; Vacik et al., 2011; Pezzatti et al., 2013). In contrast, abundant snowfall and snow cover in winter tend to prevent fires in mountain areas. According to the seasonal distribution of fire occurrence, we distinguish the following fire seasons for evaluation of indices: spring (MAM), summer (JJA) and autumn (SON). This seasonal and spatial pattern of fire occurrence is

typical for most countries located in the Alpine region (Valese et al., 2011; Vacchiano et al., 2018).



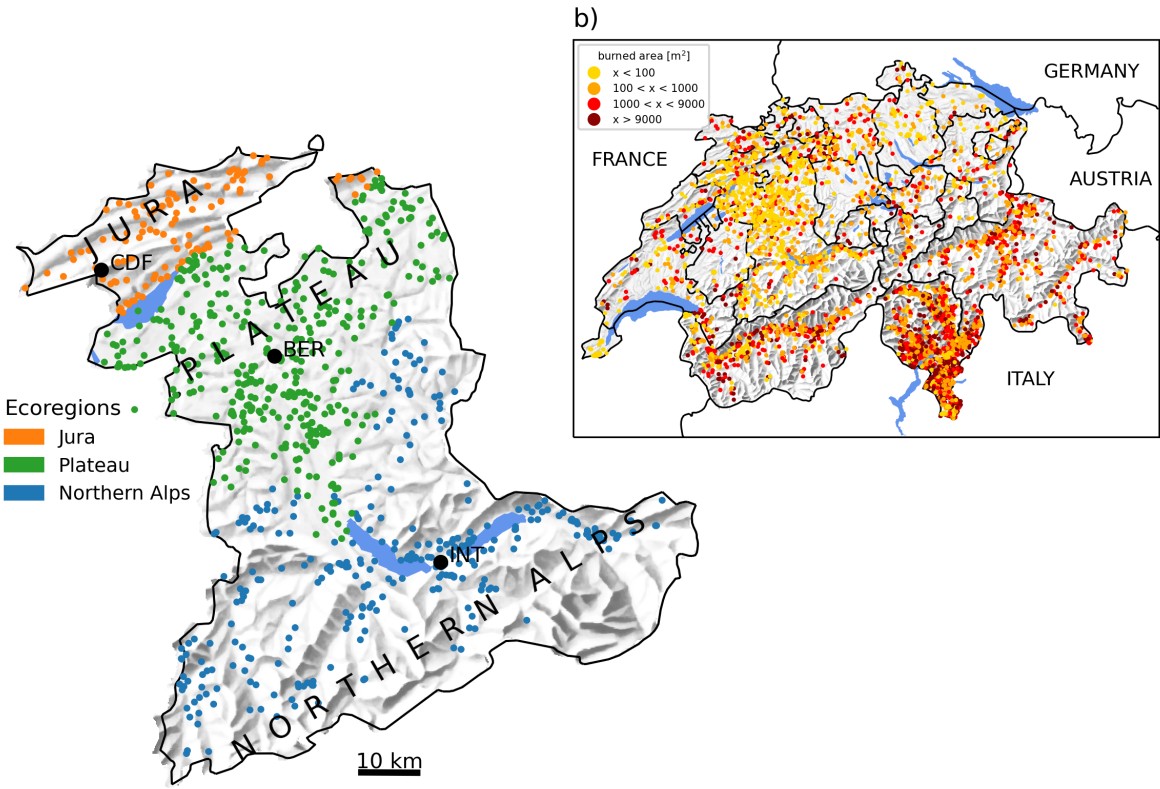

**Figure 1.** Location of fire records in the canton of Bern from 1981–2020, colored according to the ecoregions (Jura, Plateau and northern Alps). Weather stations are shown in black (CDF, BER, INT). b) Location of all fire records in Switzerland from 1981–2020, colored according to the burned area (m$^2$), with values representing the 25th, 50th, and 75th percentiles of the burned area distribution in Switzerland.

## 3.2 Assessment of fire weather indices

First, we assess the skill of various fire weather indices for the three ecoregions ( Jura, Plateau and northern Alps) and seasons (MAM, JJA and SON) using the ranked percentile score (described in Section2.3). The intercept with the 95% confidence interval of the median-based Theil-Sen model for each index is used as ranked percentile score and is shown in Figure 3. The results show that the performance of the calculated indices varies significantly by season and region. For a better overview, we will first consider each region individually and highlight the seasonal variability.

Analysis of the indices for Jura in Figure 3a shows that most indices perform well in spring (performance consistently > 0.6). The differences between the index with the best performance (FWI) and the second and third ranked indices (FFDI and FFMC) are very small and not significant (the error bars overlap). Only the DC has a score below 0.6. During summer, the CFFWIS sub-indices BUI and DC have the greatest skill with a score of 0.6, significantly outperforming the other indices. With the exception of DC, all other indices perform worse in summer than in spring. Angstroem, Neserov, and Baumgartner



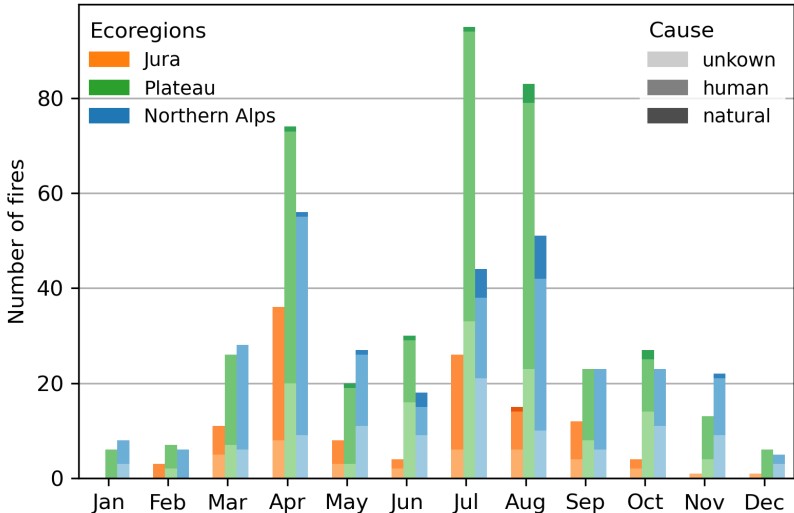

**Figure 2.** Monthly distribution of the fire records in the three ecoregions of the canton of Bern from 1981–2020, disaggregated by cause.

have the lowest skill of 0.4. During autumn, DC and BUI are again the best performing indices, with a higher intercept than in the compared to that observed in summer. The KBDI and DMC also have a good score above 0.6, and the other indices show little change in skill compared to summer. The M68dwd and Sharples show no skill in autumn ( score is zero). Notably, the 95% confidence interval of the intercept is much larger in autumn compared to the other seasons for some indices because there are fewer fire records in autumn to fit the Theil-Sen model. Comparing the mean intercept per season averaged across all indices, the indices in Jura generally perform better in spring, followed by autumn and summer. In summer and autumn, the DC and BUI outperform the other indices. In spring, the other indices (e.g., FWI, FFDI and FFMC) perform somewhat better than DC. In the Plateau ( Figure 3b ), we observe a similar seasonal performance of the indices as in the Jura. During spring, the CFFWIS sub-indices FFMC, ISI and FWI, as well as FFDI and M68dwd, show the best performance with a score around 0.6, while DC and BUI show the lowest performance. During summer, almost all indices perform somewhat worse than in spring, except BUI and DC. BUI has the best score, but does not significantly outperform other indices (FWI) or the daily T and daily H. Autumn shows the largest difference between the indices and again larger uncertainty for the estimated intercepts, with good performance by DC and BUI, followed by DMC and KBDI. The M68dwd index again shows no skill in autumn ( score is zero). In the northern Alps ( Figure 3c ), the CFFWIS sub-indices FFMC, FWI and ISI together with Neserov, M68dwd and FFDI show the best and similar results during spring, while DC shows the worst results. During summer, the BUI has the highest score of all indices, but is significantly outperformed by the daily T. During autumn, sub-indices from CFFWIS (FFMC, ISI, BUI and FWI) and FFDI perform slightly better than the other indices. Again,Sharples and M68dwd show almost no skill during autumn. Interestingly, DC does not perform well in autumn, which is in contrast to the other two ecoregions.

The results show that some indices predict fire occurrence better than others, although the differences between indices are often small and not significant, especially during the two peak fire seasons in spring and summer. Depending on the region and





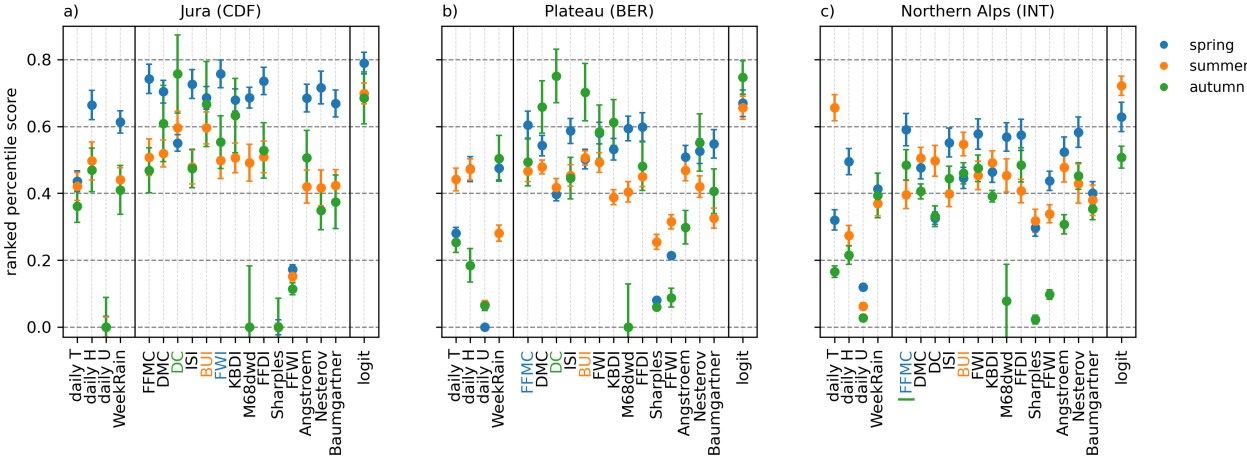

**Figure 3.** Ranked percentile score during fire days (intercept of the Theil–Sen regression model) for different fire weather indices, meteorological variables and the logit model for the three ecoregions a) Jura, b) Plateau and c) northern Alps. The colors indicate the season. The error bars show the 95% confidence interval of the intercept. The names of the indices with the highest score in each ecoregion and season are colored. The results of the area under the receiver operating characteristic curve (AUC) are shown in Figure S1 in the Supplement.

season, different indices represent fire occurence particularly well. Although the ranking of fire indices was not identical in all regions, some robust patterns emerge in the three ecoregions. The names of the indices with the highest score in each ecoregion and season are colored in Figure 3: FFMC and FWI in spring, BUI in summer, and DC (except for northern Alps) in autumn, all indices from the CFFWIS.

### 3.3 Meteorological variables and logistic regression

By including the raw meteorological variables in the assessment in Figure 3, we can examine their relative importance. Comparing the ranked percentile score of the meteorological variables shows that daily T, daily H, and WeekRain are the most important variables, while wind speed has little to no importance. However, their importance varies between seasons and regions, highlighting the very heterogeneous topographic and fuel conditions in the canton of Bern. Interestingly, the daily T has the best score for the northern Alps during the summer period (Figure 3c), clearly outperforming all fire weather indices. The daily H and WeekRain are also good indicators for fire danger during spring in all regions. Probability density functions between fire and non-fire days in Supplement Figure S2 show that daily H and WeekRain are significantly lower, and daily T is significantly higher on fire days.

Calculating fire weather indices requires specific meteorological data. For example, the sub-indices of the CFFWIS require meteorological data recorded at noon (Wotton et al., 2009), and using daily means for their calculation introduces systematic biases (Herrera et al., 2013). However, the required meteorological data are not always available, as is the case for long-term weather reconstructions dating back long periods, and for future climate projections, data are sometimes only available on a





daily temporal resolution (e.g., Pfister et al., 2020; NCCS, 2021). Therefore, we also compare the performance of a logistic

regression model (logit) that uses only the daily means as input, rather than noon values. Note that we also fitted a logit model with noon values or a combination of indices, but the performances are similar to the logit with daily means presented here. Comparing the ranked percentile score between the logit and fire weather indices in Figure 3, the logit consistently scores higher than most indices and is significantly better in summer in all three ecoregions. During spring and autumn, the logit scores better than or equal to the best performing index. Probability density functions (Figure S2 in the Supplement) for the

logit model yield the clearest distinction between fire and non-fire days and better predictive power for fire occurrence than the best performing indices. The logit model is trained on local fire statistics, which are influenced by both meteorological and non-meteorological factors such as human activity, topography and fuel types. Therefore, the logit model indirectly learns these relationships and consequently has better predictive power for identifying fire days while using only daily means as input. This illustrates the potential of statistical models that could be further improved by combining different fire weather indices

with non-meteorological factors, which was demonstrated for the fire-prone canton Ticino in Southern Switzerland (Reineking et al., 2010; Angelis et al., 2015).

## 3.4 Temporal evolution and extreme fire weather

Research indicates a general increase in fire frequency and burned area, as well as a general extension of the fire season in different regions of the Alps (Zumbrunnen et al., 2011, 2012). The high year-to-year variability of fire occurrence makes fire

analysis challenging and masks climate change signals (Monzón-Alvarado et al., 2014). We examine the temporal relationships between fire occurrence and fire weather indices to see how well the best performing indices capture interannual variability in fire occurrence. We perform this analysis for the three ecoregions and compare the number of fire days recorded annually to the annual number of extreme fire days. We define an extreme index day when the daily values of the best performing index (FFMC in spring, BUI in summer and DC in autumn) are above the 95th percentile of their 1981–2020 seasonal climatology.

The temporal evolution of recorded fire days (dashed black line) and extreme index (blue) and logit (orange) days are shown in Figure 4. The number of recorded fire days varies considerably from year to year, especially in the Plateau (Figure 4b), where the number ranges from 0 to more than 25 fire days in exceptionally hot and dry summers such as 2003 and 2015. Recent summers (2017–2020) also experienced heatwaves that caused a high number of fire days. There are years when the peaks in the number of fire days are matched by extreme index days (e.g., 2003, 2015 and 2018), but there are also years with no or few

fire days when fire danger (large number of extreme index/logit days) was elevated (e.g., 1985, early 1990s, 1998). In the Jura, fire days are generally overestimated by the index and logit model. In the Plateau, unprecedented fire days in 2003 and 2015 are fairly well reproduced by both the index and logit model, but they are overestimated in years before 2000 and underestimated in recent years (2019–2020). In the northern Alps, fire days are generally overestimated, especially by the extreme index days. There is evidence that index and logit have a positive relationship with the number of fire days. This is evidenced by a moderate

Spearman rank correlation (p ≈ 0.4) for index and a moderate to strong correlation (p ≈ 0.6 and above) for logit. Logit has a higher correlation than index, except in Jura, where the extreme index days have a strong correlation (p = 0.77). Although the vast majority of fires (>95%) in this study area are caused by human activities, the number of fires days strongly depends



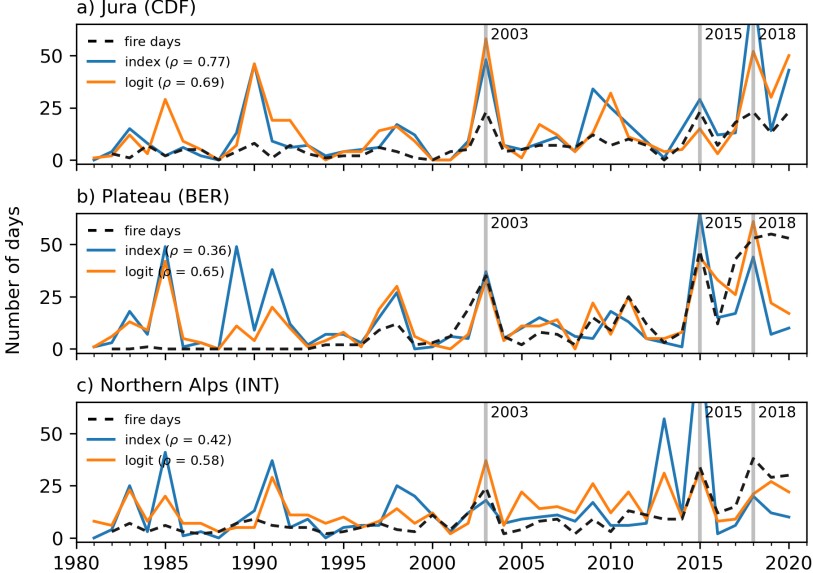

**Figure 4.** Time series of yearly number of recorded fire days (black dashed lines) and yearly number of extreme index days (blue lines) and extreme logit days (orange lines) for ecoregions a) Jura, b) Plateau and c) northern Alps from 1981–2020. The legend shows Spearman rank correlation between the number of fire days and extreme days. In all cases, an extreme day is defined as when the index (or logit) value is above the 95th percentile. For extreme index days, the index with the highest score for each season and ecoregion was used, as highlighted in Figure 3).

on favorable meteorological conditions and weather is an important driver of increased fire occurrence. A slight increase in fire days and extreme index days has been observed in recent years, but the long-term trend is weak due to large year-to-year

variability and possible asymmetry in the fire database with missing records before 2003. A slight increase in fire danger in the northern Alps is consistent with the results of other studies (e.g., Wastl et al., 2012).

## 4    Discussion

### 4.1    Seasonal and regional differences

The analysis of the indices for the canton of Bern shows some interesting features. The results presented in Figure 3 highlight

that the performance of fire weather indices and meteorological variables varies considerably by region and season. Seasonal differences are more pronounced than regional differences, which means that in most cases the indices show similar seasonal variability in all three ecoregions.

Indices such as the FFMC, FWI, and also ISI, FFDI and M68dwd are good indicators of the fire danger in spring. In summer and autumn, the BUI and DC perform best. While the final CFFWIS index FWI performs overall well in all regions and





seasons, it is not surprising that several sub-indices of the CFFWIS are better at predicting fires at certain times of the year than the final FWI index and perform better compared to other indices. This is consistent with Arpaci et al. (2013) for Austria and de Jong et al. (2016) for the UK, and can be explained by how fire behaviour changes from spring to summer due to the soil layers that appear to be most relevant for fire potential. The sub-indices of CFFWIS represent specific layer depths of the forest soil, and differences between indices can be explained by drying mechanisms and response times to weather situations (Wastl

et al., 2012). During spring, when leaf unfolding has not yet started, e.g., solar radiation reaches the forest floor, and with lots of dry litter on the floor, fires are characterized by rapidly spreading surface fires (Zumbrunnen et al., 2011) that depend on the quick-drying fine fuel in the top-most litter layer - factors reflected in the fast-responding FFMC. The moisture content of this litter responds readily to changes in atmospheric conditions. The moisture content of slow-drying fuels in deeper layers, reflected in the DC and BUI, is still high and therefore the DC and BUI perform worse. Consequently, indices describing rapid

changes in weather conditions perform well in spring. In summer, fires often occur during and after prolonged dry and hot periods when the moisture of the slow-drying fuel in the deeper layers can get low. Summers in Bern are characterized by extended dry periods and short intervals of convective precipitation (Isotta et al., 2014), that are often localized and do not reflect drought conditions at the regional scale. During this situation, the cumulative and slower responding indices (DC and BUI), i.e. they accumulate past weather patterns, can peak and their performance improves, while the performance of FFMC,

ISI, M68dwd and FFDI decreases compared to spring. Nevertheless, it is interesting to see that BUI is the best index for fire occurrence in summer, because it represents the amount of fuel available and this describes fire behaviour/severity rather than occurrence (Van Wagner and Pickett, 1985). In autumn, the DC performs better in Jura and Plateau than in northern Alps, most likely due to the increased availability of slow-drying fuel in the broadleaved forests of the lower regions. The overall poor performance of M68dwd (Käse, 1969) in autumn was also observed in Arpaci et al. (2013). M68dwd requires a snow layer

as input, which we did not provide. This could be problematic in late autumn, especially at higher latitudes. Similar to Arpaci et al. (2013), the daily temperature is the best index in northern Alps in summer.

### 4.2    Role of meteorological variables

In summer and spring, meteorological variables such as temperature, relative humidity and rain alone are well able to distinguish between fire days and non-fire days. This was also discussed in Padilla et al. (2011); Holsten et al. (2013); Arpaci et al.

(2013), which found that temperature or humidity outperformed other fire weather indices in some regions. Initially, it seems surprising that the raw meteorological inputs perform so well. To some extent, the good performance of temperature and humidity is mediated by a relationship between hot and dry weather and human activities. High daily temperatures dry fine fuels and increase flammability, while also leading to more outdoor recreation (human ignition source). Our analysis did not identify wind speed as an important factor in fire occurrence. The weather in the Alpine valleys is often influenced by dry foehn winds

from the south, which cause a significant decrease in relative humidity (Richner and Hächler, 2013), dry out the soil quickly, and thus increase the risk of forest fires (Sharples et al., 2010; Wastl et al., 2013). Zumbrunnen et al. (2009); Wastl et al. (2013) noted a high number of fires during dry foehn conditions in many regions of the Alps. The poor relationship between wind





and fire occurrence found in our study may be due to the use of daily averages, which are inappropriate for representing highly irregular foehn winds.

Nevertheless, indices combine several meteorological variables into one index that can be interpreted not only in terms of ignition but also in terms of potential fire behaviour (Wotton et al., 2009), and are therefore preferred over raw meteorological variables.

### 4.3  Potential applications and limitations

This assessment of indices has several potential applications. First, by identifying the best performing index for each ecoregion
and season, a fire danger rating system consisting of multiple, seasonally and regionally varying indices could be implemented for real-time prediction using meteorological forecasts, which is also proposed by Reineking et al. (2010) for the southern Canton Ticino. This could improve the risk assessment compared to using only one index in all seasons and regions. Second, a statistical model that translates a series of meteorological daily means into a probability of fire occurrence can serve as a basin for prediction and analyzing long-term weather reconstructions or simulate future changes to fire based on future
climate projection (the focus of a next study). Such projections will contribute to a better understanding of the spatio-temporal variability and future changes and provide decision makers with the information they need to successfully adapt to climate change.

However, this study is subject to several caveats. First, fire weather indices are based solely on weather and ignore human activities (ignition and suppression). Weather mainly provides information on fire disposition. In Switzerland, fire occurrence is
mainly caused by human activities (Conedera and Tinner, 2010) and is therefore difficult to predict by meteorological forecasts alone. Indeed, ranked percentile scores reached in Figure 3 are significantly lower than for a perfect score. Nevertheless, we see that in years with exceptionally high numbers of wildfires, e.g., 2003 and 2015, meteorological conditions were favorable, because without warm and dry weather it is usually not possible for numerous fires to occur. The moderate to strong correlations between indices and fire days underline the meteorological importance for fires. Including human factors related to ignition as
well as other social-economic (e.g., distance to roads) and environmental factors (topography, forest composition) into a fire danger model can improve performance (e.g., Reineking et al., 2010). Angelis et al. (2015) developed a statistical model for Southern Switzerland by combining the most promising fire weather indices that significantly improved wildfire prediction. Such statistical models trained on local fire statistics indirectly incorporate non-meteorological factors while still using only meteorological information as input. Here, we demonstrate that a logistic regression model has higher scores than most indices
in almost all regions and seasons (Figure 3). We present only daily means in the logit model to examine the relative importance of meteorological variables and to allow comparison with fire weather indices. Second, we validated the database records for Bern only back to 2003 using the MODIS fire product (Giglio et al., 2016). Therefore, we must assume an asymmetry in fire records with a higher percentage of unreported fires prior to 2003. In Figure 4 we see that the number of recorded fire days is very low in the years before 2000, despite high extreme fire danger. We used all recorded fires, including very small fires
and fires without size information, for validation. We hypothesize that small fires, which are either small because they are spotted quickly or have not been able to spread because of unfavorable conditions, have a weaker relationship to weather than



larger fires. Arpaci et al. (2013), who considered also only larger fires in their assessment of fire weather indices in Austria, demonstrated an overall better score compared to all fire sizes. However, using only larger fires results in a very limited number of fire records in our study area, which is a major limitation for the development of the statistical model (Angelis et al., 2015)

and fitting the logistic regression model. Third, our validation relies on individual weather stations to estimate fire danger for a larger region. While this practice is commonly used (e.g. Andrews et al., 2003; Pezzatti et al., 2020), weather stations are not necessarily located near the fire site and therefore don't have the same micro climate. This is especially true for the complex and diverse landscape of Switzerland with its steep topography and mixed forest types, and also due to local wind systems and localized convective precipitation during summers that may or may not hit a weather station. In addition, weather stations

are located in open areas and may not present fuel moisture in forests. This could be improved by the availability of spatially explicit weather information, for example, gridded fields of high-resolution reanalysis datasets (Vitolo et al., 2020).

## 5   Conclusions

The aim of this study is to evaluate and compare different fire weather indices, four meteorological variables and a logistic regression model in terms of their predictive power for fire occurrence/ignition in Northern Switzerland on a daily basis. The

study area was the canton of Bern, which represents the three different biogeographical ecoregions (Jura, Plateau and northern Alps) of Northern Switzerland. Bern is currently a low fire risk region, but anthropogenic climate change is expected to increase fire risk. (Pezzatti et al., 2016). Therefore, a better understanding of fire occurrence and its meteorological drivers is needed. We evaluate and identify the best performing index for each ecoregion and season using the ranked percentile curve (Eastaugh et al., 2012) and use historical fire records from 1981-2020 as a validation dataset (fire days vs. non-fire days).

Our results show strong seasonal and regional differences in the performance of the indices, but some robust patterns also emerge. The FFMC, FWI, and ISI, which describe short-term weather conditions and litter layer moisture content, perform best in spring. In summer and autumn, the BUI and DC, which accumulate past weather and describe moisture in deeper organic layers, have a chance to peak during prolonged drought and heat and describe fire occurrence well. All of the best performing indices are from the Canadian fire weather index system (Van Wagner and Pickett, 1985). The differences between the best

performing index and other indices are sometimes very small and not significant. As for the meteorological variables, temperature and relative humidity are the most important input variables and even outperform the indices in the mountainous region in summer. These results are consistent with those of Padilla et al. (2011); Holsten et al. (2013); Arpaci et al. (2013) and not surprising, given that fire weather indices have been developed under specific environmental conditions in other regions. This is also evident in the development of a logistic regression model with daily means as input, which shows better performance

than the indices across all ecoregions and seasons. These seasonal and regional differences motivate the use of multiple fire weather indices rather than a single, stand-alone index (Reineking et al., 2010) or a statistical model trained on local historical fire statistics (Angelis et al., 2015) to improve fire hazard prediction in the study area

The study shows that weather-derived fire prediction is challenging. Other non-meteorological factors that we have not considered here play a critical role in fire occurrence, particularly human activities to ignite but also to prevent fires. A first step



in overcoming these limitations is to combine fire weather indices with non-meteorological factors such as forest composi-
tion, topography, and human activities (Reineking et al., 2010). In addition, fire weather indices derived from high-resolution
gridded fields of regional weather models can resolve local weather conditions and contribute to a better understanding of the
spatiotemporal variability of fire risk, especially in Switzerland with its complex and diverse topography. This was recently
implemented by the European Centre for Medium-Range Weather Forecasts for their global weather forecast model (Giuseppe

et al., 2016). Nevertheless, the analysis shows that the occurrence of above-average forest fire activity in exceptionally hot and
dry summers such as 2003 and 2015 is significantly related to weather, although other factors such as fuel availability or fire
suppression also play an important role.

*Code and data availability.* The scripts used for this study are publicly available at: https://github.com/steidani/FireDanger. Meteorological
data can be downloaded from the Swiss Federal Office of Meteorology and Climatology MeteoSwiss: https://gate.meteoswiss.ch/idaweb.

Fire records from the Swissfire database were provided by WSL: www.wsl.ch/swissfire.





*Author contributions.* DS performed the data analysis and visualizations and drafted the paper. All authors commented on the paper.

*Competing interests.* The authors declare that they have no conflict of interests

*Acknowledgements.* We would like to thank Boris Pezzatti and Marco Conedera (both from WSL) for providing the fire records from the Swissfire Database, and MeteoSwiss for the weather station data. DS acknowledges funding from the Wyss Academy for Nature.



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
