# Peer review of "Assessing the performance of various fire weather indices for wildfire occurrence in Northern Switzerland"

_EGUsphere, 2022_

## Author Comment (AC1)

**Response to reviewer comments for**
**Assessing the performance of various fire weather indices for wildfire occurrence in Northern Switzerland**

Daniel Steinfield, Adrian Peter, Olivia Martius, Stefan Brönnimann

September 1, 2022

Dear Editor and Reviewers,

We thank both reviewers for their effort in reviewing our manuscript and for their detailed and critical feedback. Both reviewers appreciated our study on the evaluation of fire weather indices, but the overall feedback is rather negative and they conclude that the study lacks innovation. Although we believe the results of the study are important and novel, particularly how the performance of the indices varies seasonally and regionally across a small geographic region, we accept the criticism and therefore will not submit a revised manuscript to NHESS.

Nevertheless, we would like to comment on the points raised by the reviewers and indicate how we will further improve the manuscript for future submission.

**Lack of novelty**

Both reviewers conclude that the ideas and methods used are not innovative and the results are not of interest to a wider audience due to the small size of our study area.

This study was designed together with the Office for Forests and Natural Hazards of the Canton of Bern with a practical aim, namely to assess how well the currently operationally used indices of the Canadian system (Van Wagner and Pickett, 1985) perform in comparison to other indices. It is hence a stakeholder driven research project embedded in and guided by an operational warning framework, which leads to different research questions than a research project that is purely knowledge driven. Although the Canton of Bern is relatively small compared to other regions (in other countries), it is an interesting study region because of its diverse topography with three distinct ecoregions (the mountainous Alps and Jura, and in-between the hilly Plateau). We find that the performance of the indices varies seasonally and regionally. We believe that this result is novel and interesting enough, as it shows the advantage of using multiple indices instead of a single stand-alone index (e.g., the final FWI) to estimate fire weather danger. We decided to use an existing validation metric, the percentile score, because it was applied in previous studies (Arpaci et al., 2013; Eastaugh and Hasenauer, 2014; de Jong et al., 2016) and allows us to compare our results with theirs. This method compares indices without depending on subjectivity of threshold selection for fire danger categories, and the advantages

of the percentile score over other validation metrics have been described in Eastaugh et al. (2012). Reviewer 1 asked if we adapted the indices: It is true that one could, for example, adjust the thresholds for danger categories based on percentiles, as was done in (de Jong et al., 2016). However, this is not necessary when using the percentile score, since it compares the distributions of an index between fire and non-fire days.

In the revised version, we will improve the text by clearly describing the practical focus of this study with a new title: Assessing the performance of various fire weather indices for a pre-Alpine region in Switzerland.

**Incorrect Terminology and phrasing**

Both reviewers mention that we falsely used terms and there are unnecessary sentences.

We apologise that we were too sloppy with the terminology, and falsely used terms like "danger", "hazard" and "risk" as synonym. As pointed out, this study assesses fire weather indices that estimate fire-prone weather conditions, that is highlighting fire danger purely from a meteorological perspective. In the revised version, we will address this issue and will use a more appropriate terminology and remove unnecessary sentences. We will also clearly state that this study focuses on the Canton of Bern and was not applied to other regions. The objectives, methods and results are discussed more carefully.

**Methodology and validation metric**

Revierwer 1 criticises that the data is insufficient and the methodology is inadequate.

We do not quite agree with that.

Let us start with the data (weather station and fire records): It is a common practice to calculate indices based on weather stations that represent fire records in a larger region. This has been done in (Angelis et al., 2015; Pezzatti et al., 2020) for similar or larger regions, for example. We selected our three weather stations because they represent the climatology of the three ecoregions and they provide 40-years of hourly measurements for 2m temperature, 2m humidity, 10m wind speed and precipitation. We agree that with only one weather station per ecoregion, we may miss very localized weather phenomena such as thunderstorms and wind systems like foehn. We discuss this issue in the manuscript and find it somewhat unfair that reviewer 1 uses our argument against us, as it would also apply to the studies mentioned above. To better address this issue in the revised manuscript, we will use additional weather stations to test the robustness of the results with respect to the choice of weather station, and we will calculate mean meteorological values for each ecoregion to calculate the indices. However, not all weather stations provide long-term measurements of all four meteorological input variables, so we must focus on a shorter time period, reducing the number of fire records. Fire records in the canton of Bern are considered nearly complete thanks to intensive retrospective source analysis (Pezzatti et al., 2016). In retrospect, the use of MODIS as an additional source for the fire records in the canton of Bern is not necessary and is not considered in the revised version. We chose not to include fire records from neighboring cantons because of large inconsistencies between the cantons in how fires are recorded in the Swissfire database. This inconsistencies in fire records

between cantons can be seen in Figure 1b in the manuscript).

About the verification metric: The percentile score was applied in previous studies to assess the performance of fire weather indices. It compares the values of an index during fire and non-fire days and is therefore not depending on subjectivity of threshold selection for fire danger categories. In addition, we construct and show 95% confidence intervals around the scores in Figure 3 to demonstrate statistical significance (which is equivalent to a significance level or p-value of 5%). To be able to calculate statistical significance, we needed enough fire records. This is the reason why we used a long time period from 1981 to 2020 and didn't focus on large fires or only forest fires. For robustness, we also calculated the more known AUC ROC metric (see Supplement) with similar results to the percentile score.

**Structure of the paper**

Both reviewers point out that the paper is not well structured. We would like to improve the structure and readability of the paper as follows: We will remove the additional analyses in Section 3.3 (logistic regression model) and Section 3.4 (Temporal evolution and extreme fire weather), focus on the assessment of the indices, and the objectives, methods and results are discussed more carefully.

**References**

Angelis, A. D., Ricotta, C., Conedera, M., and Pezzatti, G. B.: Modelling the Meteorological Forest Fire Niche in Heterogeneous Pyrologic Conditions, PLOS ONE, 10, e0116 875, https://doi.org/10.1371/journal.pone.0116875, 2015.

Arpaci, A., Eastaugh, C. S., and Vacik, H.: Selecting the Best Performing Fire Weather Indices for Austrian Ecoregions, Theoretical and Applied Climatology, 114, 393–406, https://doi.org/10.1007/s00704-013-0839-7, 2013.

de Jong, M. C., Wooster, M. J., Kitchen, K., Manley, C., Gazzard, R., and McCall, F. F.: Calibration and Evaluation of the Canadian Forest Fire Weather Index (FWI) System for Improved Wildland Fire Danger Rating in the United Kingdom, Natural Hazards and Earth System Sciences, 16, 1217–1237, https://doi.org/10.5194/nhess-16-1217-2016, 2016.

Eastaugh, C. S. and Hasenauer, H.: Deriving Forest Fire Ignition Risk with Biogeochemical Process Modelling, Environmental Modelling & Software, 55, 132–142, https://doi.org/10.1016/j.envsoft.2014.01.018, 2014.

Eastaugh, C. S., Arpaci, A., and Vacik, H.: A Cautionary Note Regarding Comparisons of Fire Danger Indices, Natural Hazards and Earth System Sciences, 12, 927–934, https://doi.org/10.5194/nhess-12-927-2012, 2012.

Pezzatti, G. B., De Angelis, A., and Conedera, M.: Potenzielle Entwicklung der Waldbrandgefahr im Klimawandel, in: Wald im Klimawandel. Grundlagen für Adaptationsstrategien, pp. 223–245, Haupt, Bern, 2016.

Pezzatti, G. B., De Angelis, A., Bekar, İ., Ricotta, C., Bajocco, S., and Conedera, M.: Complementing Daily Fire-Danger Assessment Using a Novel Metric Based on Burnt Area Ranking, Agricultural and Forest Meteorology, 295, 108 172, https://doi.org/10.1016/j.agrformet.2020.108172, 2020.

Van Wagner, C. E. and Pickett, T. L.: Equations and FORTRAN Program for the Canadian Forest Fire Weather Index System, Tech. rep., Canadian Forestry Service, 1985.